# Lupus Nephritis Biomarkers: A Critical Review

**DOI:** 10.3390/ijms25020805

**Published:** 2024-01-09

**Authors:** Fatima K. Alduraibi, George C. Tsokos

**Affiliations:** 1Department of Medicine, Division of Clinical Immunology and Rheumatology, Beth Israel Deaconess Medical Center, Harvard Teaching Hospital, Boston, MA 02215, USA; 2Department of Medicine, Division of Clinical Immunology and Rheumatology, University of Alabama at Birmingham, Birmingham, AL 35294, USA; 3Department of Medicine, Division of Clinical Immunology and Rheumatology, King Faisal Specialist Hospital and Research Center, Riyadh 11564, Saudi Arabia

**Keywords:** biomarker, systemic lupus erythematosus, lupus nephritis

## Abstract

Lupus nephritis (LN), a major complication in individuals diagnosed with systemic lupus erythematosus, substantially increases morbidity and mortality. Despite marked improvements in the survival of patients with severe LN over the past 50 years, complete clinical remission after immunosuppressive therapy is achieved in only half of the patients. Therefore, timely detection of LN is vital for initiating prompt therapeutic interventions and improving patient outcomes. Biomarkers have emerged as valuable tools for LN detection and monitoring; however, the complex role of these biomarkers in LN pathogenesis remains unclear. Renal biopsy remains the gold standard for the identification of the histological phenotypes of LN and guides disease management. However, the molecular pathophysiology of specific renal lesions remains poorly understood. In this review, we provide a critical, up-to-date overview of the latest developments in the field of LN biomarkers.

## 1. Introduction

Systemic lupus erythematosus (SLE) is a chronic systemic autoimmune disease characterized by the presence of autoantibodies (autoAbs), autoreactive B and T cells, and the dysregulation of cytokines, which lead to inflammation and cause damage to multiple organs [1,2,3,4]. The prevalence of SLE in the United States ranges from 20 to 150 cases per 100,000 people [5,6,7,8]. The etiology and pathogenesis of SLE are not well understood; the factors that lead to disease onset are highly variable, and the disease manifests systemically with manifestations resulting from the injury of multiple tissues. The kidney is the most commonly involved organ in this disease and contributes extensively to morbidity and mortality [9,10,11]. 

Lupus nephritis (LN) has been classified histologically into six types, which are determined by the location and the type of histological changes. There is variability among the six classes in terms of response to treatment and preservation of the kidney function and the development of end-stage disease [12]. LN is characterized by inflammation of the kidney arising from complex interactions between the innate and adaptive immune responses and the kidney parenchyma [13,14]. Extensive research has recognized the contributions of genetic, epigenetic, and environmental factors to LN; however, the precise etiology and pathogenesis of LN remain unclear. The onset and progression of LN, as well as the response to treatment, are highly variable among patients, highlighting the need to develop biomarkers to assess these diverse aspects of this disease [15]. The pathogenesis involves several genetic variants, including those that hinder the efficient elimination of dying cells, resulting in the persistence of nuclear antigens in the extracellular space. The released autoantigens act as triggers for innate and adaptive immune responses, while other genetic variants influence the magnitude of these immune reactions and the proper function of immune tolerance checkpoints, ultimately promoting the expansion of autoreactive T and B lymphocytes [15,16]. Consequently, circulating antinuclear antibodies, along with other autoAbs, lead to the formation of ICs that propagate systemic inflammation, leading to organ damage. The deposition of ICs in the renal microvasculature triggers inflammation and organ injury, accounting for the extensive morbidity and mortality associated with the disease [17]. The persistence of autoAbs amplifies systemic autoimmunity and exacerbates LN by facilitating an inflammatory response. These events may accentuate autoantigen presentation and induce immune responses specific to kidney-specific autoantigens [18]. Furthermore, ICs, local complement system activation, and the recruitment of immune cells cumulatively contribute to kidney damage. In response to autoantigens, immune cell memory may introduce further complications in the context of SLE, LN disease progression, and flares. The persistence of memory T cells and plasma cells in the bone marrow and other lymphoid organs makes them less susceptible to traditional therapeutic interventions [19,20,21,22,23].

In recent years, remarkable progress has been made in understanding the pathophysiology of LN, leading to the identification of novel biomarkers related to cellular and inflammatory mechanisms. For instance, advanced techniques, such as single-cell (sc) RNA sequencing, have offered valuable insights into the intricate composition of immune cell populations in the kidney tissues of patients with LN [13]. As LN can be categorized into different types, there is the possibility that certain biomarkers correlate with specific types of LN. For instance, the signature of type I interferon (IFN) is predominantly observed in the kidneys and skin of individuals diagnosed with proliferative LN [3,24], whereas patients with membranous LN exhibit different transcriptomic patterns [24], and transcriptome-based investigations using animal models have provided insights into the progression and phases of LN development [25].

Recently, there has been increasing interest in the development of early prediction and detection methods to prevent the onset, manage relapses, and mitigate the complications associated with chronic kidney disease (CKD) as alternatives to renal biopsy. Although invasive, the current diagnostic approach for LN typically involves renal biopsy, which is considered the gold standard for establishing an initial diagnosis. However, it poses challenges regarding the frequent monitoring of disease progression and treatment response.

Currently, the scientific community is building momentum to incorporate more comprehensive approaches, including proteomics and metabolomics, to develop reliable diagnostic and prognostic markers. Here, we review various published approaches regarding the development of biomarkers for LN.

## 2. LN Biomarkers

### 2.1. Serum AutoAbs

AutoAbs have emerged as invaluable biomarkers that provide insights into disease activity and progression. Several autoAbs have been found to be associated with LN [13]. Associations between specific autoAbs, including antibodies against ds DNA (dsDNA) and -C1q, and distinct histological classifications of LN have been reported [26,27,28]. Anti-phospholipid antibodies are also associated with renal thrombotic microangiopathy [29,30]. Crescent formation in the kidney often indicates the presence of antibodies that damage the renal capillaries [30,31,32].

Anti-dsDNA antibodies and C1q are frequently used as LN biomarkers and have been endorsed by guidelines for disease monitoring [33]. The presence of antibodies against C1q is associated with LN [34], with high specificity (92%) and sensitivity (56%) [35]. This suggests that the level of antibodies against C1q can serve as a valuable diagnostic tool and help distinguish LN cases from those of other nephritides [35]. Additionally, antibodies against C-reactive protein (CRP) have been detected in patients with active LN, and their levels were found to be correlated with the Systemic Lupus Erythematosus Disease Activity Index (SLEDAI) score (*p* = 0.002) [36]. Moreover, high baseline levels of antibodies against CRP significantly predict an unfavorable outcome (*p* = 0.021) during the second year of therapy [36]. This finding highlights the potential of using antibodies against CRP as valuable biomarkers of disease activity in LN.

Antibodies against α-enolase (anti-ENO-1) are formed when α-enolase externalizes during NETosis [37]. Their presence has demonstrated significant potential in predicting LN in patients with SLE, with an area under the curve (AUC) of 0.81 and a *p*-value of 0.001 [38]. Furthermore, an increased prevalence of the anti-ENO-1 IgG2 isotype discriminates patients with LN from those with other nephritides and non-renal SLE, with an AUC of 0.82 and a *p*-value of 0.02 [39,40].

Antibodies against alpha-actinin (AaA) have shown promise as diagnostic markers for LN. In a previous study that compared patients with LN and those with SLE without nephritis, the serum AaA levels were considerably lower in those with LN, with a sensitivity of 60%, a specificity of 90%, and a positive predictive value (PPV) of 85.7% [41]. These findings indicate that serum AaA levels may serve as a marker for distinguishing patients with LN from those with SLE without nephritis [41]. 

Histones are a primary component of the nucleosome, and studies have indicated a strong correlation between antibodies against chromatin/nucleosomes and LN as well as an increased risk of developing proliferative LN [42,43]. Histone 1 forms part of the apoptotic chromatin constituents in the dense electron deposits of the glomerular basement; these are recognized targets of nephritogenic dsDNA antibodies [44,45,46,47]. Antibodies to chromatin are present in patients with LN, but they are not particularly useful in monitoring disease activity or response to treatment [48]. 

Interestingly, numerous studies have reported that patients with LN and antineutrophil cytoplasmic antibodies (ANCA), specifically those who are anti-myeloperoxidase (MPO)/proteinase 3 (PR3)-positive or double positive for anti-MPO and anti-PR3 antibodies, exhibit serologically active SLE (higher dsDNA antibody titers and lower serum C4 concentrations) compared with ANCA-negative LN patients. Additionally, patients in the subgroup with proliferative LN display more necrosis in renal biopsy specimens than ANCA-negative patients. However, it is noteworthy that no significant difference was reported between the outcomes of these groups [32,49,50]. Yet, given the poor prognosis of ANCA-positive cases, it is suggested that all patients with LN be screened for ANCA. The measurement of autoAb levels could potentially enhance the early detection, monitoring, and management of LN. Nevertheless, rigorous prospective studies are required to establish their utility as biomarkers.

### 2.2. Cytokines/Chemokines

In addition to autoAbs, various serum proteins have emerged as potential LN biomarkers. These include cytokines, chemokines, and adhesion molecules. Cytokines regulate the immune response and are crucial in the pathogenesis of LN, particularly in attracting leukocytes and in the development of the inflammatory response [51,52]. The deposition of ICs on glomeruli initiates and activates the complement pathway, attracting immune cells and triggering the release of inflammatory cytokines, thereby exacerbating glomerular damage. For instance, B-cell activating factor (BAFF or BLyS), produced by glomerular macrophages and mesangial cells, stimulates B-cell activation both directly and by inducing the production of pro-inflammatory molecules, such as IFN-α, perpetuating a detrimental cycle in the renal microenvironment. Moreover, cytokines, such as urinary monocyte chemoattractant protein-1 (MCP-1/CCL2), BAFF, and TNF-related weak inducer of apoptosis (TWEAK), are pertinent in predicting active LN, assessing therapeutic responses, and forecasting disease flares [53,54].

Furthermore, distinct cytokines predominate in different types of LN. For example, in proliferative LN (class III/IV), the deposition of ICs beneath the endothelium leads to mesangial cell proliferation, an increase in extracellular matrix production, and the release of a remarkable array of pro-inflammatory cytokines, such as type I IFN, interleukin (IL)-1β, IL6, IL8, IL-37, and IL-17A [55]. Conversely, membranous LN (class V) is characterized by the subepithelial localization of ICs, resulting in increased complement activation and reduced inflammatory responses [55]. Levels of cytokines may prove useful in distinguishing between proliferative and non-proliferative forms of LN, and this information may be useful in guiding precise treatment and management strategies and may eliminate the need for invasive LN biopsy [55]. Several cytokines have been examined as potential biomarkers to assess the activity, severity, and renal involvement of SLE.

Chemokines are small chemotactic cytokines that typically range in size from 8 to 10 kDa and play crucial roles in regulating the migration and localization of immune cells [56]. Chemokines and their receptors have been implicated in the pathogenesis of LN in both patients with SLE and lupus-prone mice. Several chemokines have shown promising results as biomarkers for the diagnosis and prognosis of LN.

#### 2.2.1. Serum

BAFF and a proliferation-inducing ligand (APRIL) are cytokines belonging to the tumor necrosis factor (TNF) family; they act on B cells and drive the activation of these cells, thereby contributing to the pathogenesis of LN [57,58]. Reduced baseline serum levels of BAFF are predictive of clinical and pathological responses, demonstrating a 92% PPV for clinical responders in cases of proliferative LN [59]. Furthermore, the inhibition of BAFF activity with belimumab, which has been approved for treating patients with active LN, has shown promising results [60]. Serum levels of APRIL (sAPRIL) are associated with its presence in the urine and its histological activity and can help predict the likelihood of treatment failure [61]. A previous study showed that the urine levels of BAFF and APRIL were notably higher in patients with active proliferative LN than in those with SLE with no renal involvement [53]. Moreover, the AUC values of 0.825 for urine BAFF and 0.781 for urine APRIL helped in differentiating between active LN and active SLE without renal involvement [53].

IFN inducible protein-10 (IP-10 or CXCL10) is a chemokine belonging to the ELR CXC family which is secreted by immune and non-immune cells [62]. It is produced in response to IFN activation and guides lymphocytes to the affected organs in lupus-prone mice and patients with SLE [63]. The urine and serum levels of IP-10 are promising potential biomarkers of lupus activity, as they can be used to distinguish between active and inactive lupus cases [54,64]. 

The serum IL-6 levels are higher in patients with SLE than in healthy individuals and correlate with the SLEDAI (*p* = 0.018) [65]. In a previous study, despite the association of serum and urine IL-6 levels with clinical manifestations of LN, the inhibition of IL-6 activity failed to show therapeutic benefits in clinical tests [66].

Serum IL-10 levels display high accuracy in distinguishing active LN from inactive LN, as evidenced by an AUC of 0.87 (*p* = 0.003), 70.6% sensitivity, 100% specificity, and a notable correlation with the SLEDAI (*p* = 0.004) [65,67].

IL-16, produced by immune and non-immune cells [68,69], exhibits markedly increased levels in the urine of patients with active LN, particularly in proliferative cases. The observed decrease in urine IL-16 levels during therapeutic intervention may serve as a useful marker for assessing therapeutic responses in LN [70].

The serum angiopoietin-2 (Ang2) levels are higher in patients with SLE and those with LN than in healthy individuals [71]. Ang2 levels correlate with the SLEDAI, 24 h proteinuria, and histological activity, which suggests their usefulness as a biomarker for LN [71]. Nevertheless, Ang2 cannot distinguish between the types of LN lesions, particularly the proliferative and non-proliferative forms [71]. Serum angiopoietin-like protein 4 (Angptl4) and angiostatin are gaining attention as potential biomarkers for LN [51,72,73]. The serum Angptl4 levels effectively differentiated patients with active LN from those with active SLE without renal involvement and displayed a strong link with the renal SLEDAI (rSLEDAI), with an AUC of 0.96 [73]. Angiostatin has a strong association with LN, as reported by several studies (AUC = 0.95–0.99; *p* < 0.001) [51,74,75]. Its correlation with the rSLEDAI, SLEDAI, and National Institutes of Health LN index underscores its value as a primary marker for LN monitoring [74,75]. Furthermore, comprehensive proteomic analyses have focused on Angptl4 and angiostatin as valuable urine indicators for monitoring LN progression and renal histology in patients with SLE [51,72,73]. Likewise, in another study, urine angiostatin levels could not be used to discriminate between patients with LN and those with CKD, with an AUC of 0.56 [76].

Serum TNF receptor-associated factor 6 is a cytokine associated with LN activity and has shown promise as a diagnostic marker for LN [77]. Serum human epididymis protein 4 levels are high in SLE, especially in patients with LN, particularly in those with increased anti-dsDNA antibody levels and decreased C3 levels. Moreover, these levels have been shown to have a predictive value for the diagnosis of LN [78].

#### 2.2.2. Urine

MCP-1/CCL2 is a chemokine belonging to the C-C family, and its function is to recruit leukocytes. In a meta-analysis, the urine MCP-1 (uMCP-1) levels were found to have a sensitivity of 89% and a specificity of 63% in differentiating active LN from inactive disease [79]. Furthermore, its levels correlated with kidney disease activity and ongoing kidney impairment and were increased in cases of proliferative glomerulonephritis [80,81,82,83]. Interestingly, uMCP-1 has the potential to predict upcoming kidney disease flares 2–4 months in advance and to reflect the effectiveness of treatment [81,82].

Urine IL (uIL)-17 and urine transforming growth factor beta 1 (uTGF-β1) have shown promise as potential LN biomarker candidates [84,85]. A previous study reported significantly elevated levels of uTGF-β1 and uIL-17 in patients with severe LN compared to those in healthy participants (*p* < 0.05) [84]; the AUCs for uTGF-β1 and uIL-17 were 66.50% and 71.70%, respectively [84]. These findings suggest that uTGF-β1 and uIL-17 are potential indicators of disease severity and that they can be valuable in distinguishing severe LN cases from mild ones [84].

Several urine proteins have emerged as promising tools for the diagnosis and ongoing monitoring of LN [51,86]. Notably, a study showed that TWEAK had an AUC of 0.82, demonstrating its predictive efficacy in LN [54]. Furthermore, when combined with UMCP-1, TWEAK effectively distinguishes between active and inactive LN (AUC = 0.89) and predicts the progression of end-stage kidney disease (ESRD) (AUC = 0.78) [54,87,88]. Additionally, urine levels of IL-12p40, IL-15, and thymus- and activation-regulated chemokine have been reported to be higher in patients with active LN than in those with inactive SLE and healthy individuals and to be correlated with the rSLEDAI [89]. Furthermore, urine clusterin has shown promise as a marker for tubulointerstitial lesions in LN [90]. Urine osteoprotegerin levels are notably elevated in patients with active LN; they correlate with disease activity, and it has been suggested that they predict poor treatment response and LN relapse [91]. Finally, urine levels of transferrin and ceruloplasmin are elevated in patients with LN compared to those in individuals without LN [92].

### 2.3. Cell Adhesion Molecules (CAMs)

CAMs are essential for guiding leukocyte movement across the endothelium toward the sites of inflammation [93]. Existing studies have investigated the use of specific CAMs, including the activated leukocyte cell adhesion molecule (ALCAM), intercellular adhesion molecule 1 (ICAM-1), vascular cell adhesion molecule 1 (VCAM-1), neural cell adhesion molecule 1 (NCAM-1), and L-selectin, as biomarkers in LN [51,72,73,75,80,94,95,96,97,98,99].

Urine CAMs (uCAMs) have demonstrated remarkable precision in distinguishing between active and inactive LN as well as SLE without LN [95]. The discriminative power of these molecules is reflected in an AUC that consistently surpasses the 0.8 threshold, except for neutrophil gelatinase-associated lipocalin (NGAL) [100,101,102]. Moreover, a combination of urine markers (uVCAM-1, uCystatinC, and uKIM-1) yielded a promising AUC of 0.80 (95% CI: 0.69–0.90), highlighting the potential to differentiate between the proliferative and membranous forms of LN [54]. Levels of uCAMs collectively serve as robust tools in evaluating LN activity and informing clinical decisions [103]. It has been suggested that urine ALCAM (uALCAM) levels are more efficient in distinguishing proliferative LN from membranous LN [95]. Moreover, uALCAM and urine VCAM-1 (uVCAM-1) levels exhibit a strong correlation with renal histological activity, underscoring their potential value as LN activity markers [75,80,95,99]. Elevated initial levels of uALCAM and uVCAM-1 might signal the waning of renal function [94]. The incremental benefit of integrating these markers is yet to be conclusively assessed. A longitudinal study revealed an elevation in serum VCAM-1 levels approximately 4.5 months before a recorded renal flare, which decreased after treatment [103]. The urine-soluble VCAM-1 and VCAM-1 levels are higher in patients with LN than in healthy individuals [76]. Furthermore, VCAM-1 has outstanding potential as a biomarker for predicting a renal biopsy activity index score greater than 7, which is associated with a poor long-term prognosis [75]. Serum NGAL levels have also been shown to be a potential biomarker for differentiating patients with LN from those without nephritis [76]. Specifically, at onset, NGAL levels serve as the best predictor, outperforming VCAM and KIM1 levels in identifying treatment responders versus non-responders 6 months after the induction phase, registering an AUC of 0.78 [80]. Furthermore, serum NGAL levels are elevated in patients with active SLE and could be used to gauge the response to treatment [80,104,105]. Additionally, serum NGAL levels are elevated in patients with active SLE, and urine NGAL levels can serve as a predictor of the response to treatment. Serum levels of Axl can distinguish active LN from non-renal SLE and provide insights into long-term renal outcomes [106,107,108]. Finally, L-selectin levels are closely associated with LN activity and related organ damage metrics [73,75,98].

### 2.4. Other Protein/Lipid Molecules

Urine-soluble CD163 (uCD163), a transmembrane scavenger receptor, is expressed by macrophages and monocytes and is a useful biomarker with an AUC of 0.998 for differentiating between active and inactive LN [109]. In addition, uCD163 levels are indicative of clinical and histological renal activities in LN [110,111]. Furthermore, uCD163 levels measured 6 months after therapy initiation have been shown to predict kidney recovery in LN, with more than 87% accuracy [109].

Epidermal growth factor (EGF) concentrations are reduced in the urine of patients with active nephritis and are associated with severe renal outcomes, such as elevated serum creatinine levels and ESRD [112].

Ceramides, characterized by modifications of the sphingosine backbone, are emerging as potential biomarkers of LN [113,114]. These diverse lipids play crucial roles in cellular signaling. The serum levels of ceramides, such as C16cer, C18Cer, C20Cer, and C24:1Cer, are higher in patients with LN with kidney impairment than in healthy individuals and patients with SLE without kidney impairment [113]. C24:1dhCer is being recognized as a remarkable indicator of kidney impairment in patients with SLE [113].

### 2.5. Complement

The complement system plays a crucial role in the pathogenesis of SLE, and the serum levels of certain components have been used to monitor disease activity [115,116,117,118]. In a study of patients with LN who underwent a repeat renal biopsy two years later, the prolonged depression of serum C3 levels was associated with a trend toward a worsening chronicity index, whereas normalization of C3 was associated with a reduction in the activity index in the repeat biopsy [118]. Hypocomplementemia, particularly low levels of C3 and C4 alone, may not adequately reflect disease activity, as the sensitivity and specificity of C3 for SLE are 80% and 14%, respectively [117]. The levels of the C4 component breakdown product C4d have been found to be considerably increased during disease flares in patients with SLE, with a 68% PPV. C4d levels also correlate with LN, with a 79% sensitivity [119]. Additionally, the C4d/C4 ratio has been found to be more specific, sensitive, and effective in distinguishing LN from non-LN cases [120]. Complement factor H-related proteins (CFHRs), which encompass CFHR1 through CFHR5, are part of the broader factor H/CFHR family. The levels of CFHR3 and CFHR5 are associated with disease activity in LN [121].

### 2.6. MicroRNAs (miRNAs)

miRNAs have been reported to play important roles in the development of LN and kidney fibrosis [122,123]. The stability of miRNAs in body fluids makes them attractive diagnostic and prognostic biomarker candidates for human diseases [124]. Several miRNAs serve as potential disease biomarkers for LN. Epigenetic studies regarding SLE have highlighted a set of urinary miRNAs, including miR-146a, miR-204, miR-30c, miR-3201, and miR-1273e, that are associated with LN [125]. These miRNAs play roles in LN-specific pathways, such as nucleic acid processes and inflammation, as well as in kidney function regulation through WNT and TGF-β signaling. The levels of miR-146a are reduced in the sera of patients with LN and are associated with disease activity [77]. Notably, baseline miR-146a levels are associated with renal flares and ESRD progression [126]. The urine levels of miRNA-135b, which originates from tubular cells, vary between treatment responders and non-responders [127]. Circulating miR-21 levels have been reported to be markedly increased in patients with LN compared to those in healthy controls; thus, they can be used to discriminate between patients with LN and controls [128]. A combined calculated value of circulating miRNAs in plasma, including miR-125a, miR-142-3p, miR-146, and miR-155, has shown promise in distinguishing patients with LN from healthy individuals [129]. Additionally, urine exosomal miR-29c levels negatively correlate with the histological chronicity index and glomerular sclerosis. Its expression levels have a remarkable predictive value for chronicity in patients with LN [130]. In addition to miRNAs, long non-coding RNAs (lncRNAs) are also involved in LN development. The levels of lncRNA RP11-2B6.2 are usually elevated following the kidney biopsies of patients with LN and positively correlate with IFNa signature scores and disease activity [131]. Additionally, the lnc3643 levels are considerably reduced only in patients with SLE and proteinuria; accordingly, these levels can distinguish LN from SLE without nephritis [131].

### 2.7. Genetics 

Genetic susceptibility contributes to the complex etiology of LN. Variants in certain genes have been linked to worse outcomes in patients with LN. Owing to the widespread availability of whole-genome sequencing, these variants can be identified and used to predict disease outcomes. One to three percent of patients with SLE may have a single-gene (monogenic) defect, and these patients experience a high incidence of LN [132,133,134]. Such cases often involve DNA/RNA clearance (e.g., DNASE1L3), complement pathways (e.g., C1q or C4), and DNA/RNA detection, which lead to type I IFN activation (e.g., TLR7), and some LN cases were reported to harbor variants of these genes. In-depth genomic studies spanning the sporadic to polygenic manifestations of SLE have identified more than 100 susceptibility genes. These variants include the genes involved in cell death (e.g., FAS), effective IC management (e.g., FcGR), and the amplified immune responses of T and B cells [13,135]. For example, recent advances in whole-exome sequencing have revealed that novel mutations in the TNF alpha-induced protein 3 (*TNFAIP3*) gene are linked to LN. These mutations lead to increased activity in nuclear factor kappa B and type I IFN pathways, along with a rise in pro-inflammatory cytokines [136]. Specifically, this study linked three novel mutations with LN: c.634+2T>C, exon 7–8 deletion, and c.1300_1301delinsTA (p. A434*) [136]. Additionally, various databases have reported *TNFAIP3* mutations associated with kidney involvement, such as the p.Q187* mutation in a patient with proliferative LN and the p.F224Sfs*4 mutation in a patient with membranous LN [136,137,138]. Moreover, a genome-wide association study focusing on female SLE patients of European ancestry with LN has identified genes, such as *FCGR*, *STAT4*, and *BANK1*. These genes have been shown to correlate with both the occurrence and severity of LN; this has been confirmed across various cohorts [139,140,141,142,143]. Wang et al. used weighted gene co-expression network analysis to identify four hub genes, namely *CD53* (AUC = 0.995), *TGFBI* (AUC = 0.997), *MS4A6A* (AUC = 0.994), and *HERC6* (AUC = 0.999), which are involved in the development of the inflammatory response and immune activation in LN (*p* < 0.0001) [144]. Yavuz et al. discovered that *MERTK*, a novel genetic region, contributes to the risk of developing LN and ESRD in patients with SLE, a finding that has been replicated across various ethnicities [145]. MERTK, a member of the Tyro3/Axl/Mer receptor kinase family, is the primary receptor on macrophages for apoptotic cells [146,147]. It plays a key role in the regulation of the innate immune response through efferocytosis and notably influences the production of cytokines, such as IL-10, TGF-β, IL-6, and IL-12 [148,149]. Furthermore, *MERTK* is critical in suppressing TLR-mediated innate immune responses by activating STAT1, which leads to an anti-inflammatory feedback through the production of the cytokine signaling suppressors *SOCS1* and *SOCS3* [150]. Other genes, such as *PRDM1*, are associated with proliferative LN [145] and play a vital role as modulators of dendritic cell function and repressors of the IFN-β gene [151].

Moreover, an *APOL1* gene variant has been identified as an independent risk factor for faster progression to ESRD in patients with LN [152,153,154]. Although the precise mechanisms by which *APOL1* facilitates kidney disease progression have not been fully elucidated, it is believed that these variants directly affect the function and structure of kidney parenchymal cells and may exacerbate tissue inflammation, thereby influencing the severity of LN. Additionally, human leukocyte antigen (HLA) variants have long been found to be linked to the development of LN, with HLA-DR3 and HLA-DR15 increasing the risk significantly [155]. 

In terms of classifying patients based on their risk of nephritis, Chen et al. observed that a high polygenic risk score for SLE correlates with poorer prognostic factors like earlier age of onset and LN [156]. Kwon et al. reported in a Korean cohort that an individual’s highest weighted genetic risk score (wGRS), calculated from 112 well-validated non-HLA single nucleotide polymorphisms and HLA haplotypes of SLE risk loci, was independently associated with the development of LN and the production of the anti-Sm antibody, compared to those in the lowest wGRS quartile [157]. This association was observed regardless of the age of onset [157]. Webber et al. observed that both HLA and non-HLA SLE risk-weighted genetic risk scores were significantly associated with the risk of proliferative LN in two large, multi-ethnic cohorts of 1251 SLE patients [158]. This may indicate that SLE risk loci are of greater importance in the development of proliferative LN, as opposed to non-proliferative LN [158]. Despite these developments, the full spectrum of genetic risk factors for LN and its progression to ESRD still needs to be better understood and further developed. Taken together, a gene expression score, including all variants known to contribute to the development of LN (class, severity, and risk of ESRD), can be used to screen all patients with SLE, identify patients at high risk of LN, and predict various clinical outcomes in SLE patients.

### 2.8. Epigenetics

Epigenetic processes, particularly those involving DNA methylation and histone modifications, are crucial for LN development [159]. Irregularities in DNA methylation patterns have been observed in genes linked to immune responses and inflammatory processes in patients with LN, which indicates their potential contribution to disease development [159]. For example, alterations in the DNA methylation of the *MERTK* gene may modify its activity and increase the risk of SLE-ESRD [145]. Similarly, histone modifications can influence gene expression profiles in immune cells, potentially contributing to the dysregulation of immune responses observed in LN [160]. New technologies that are used to assess the epigenome of immune cells may evolve into useful diagnostic and disease-monitoring tools. 

## 3. Conclusions and Future Directions

The complexity of LN pathogenesis poses challenges for the development of biomarkers for monitoring disease activity in response to treatment. The biomarkers discussed in this review represent elements that reflect multiple aspects of the immune response as well as aspects of vascular and parenchymal cell injury (Table 1) [161,162,163,164,165,166,167,168,169,170,171,172,173,174,175,176,177,178,179,180,181,182,183,184,185,186,187,188,189,190,191,192,193,194,195,196,197,198,199,200,201,202,203,204,205,206,207,208]. It is obvious that with each biomarker in patients with LN, variable pathogenetic pathways are involved, bespeaking the heterogeneity of the disease. Although tools that integrate multiple proteins, or other molecules, may have a greater value as biomarkers, the same limitations may apply. Although in some patients the diagnosis of SLE coincides with that of LN, LN may develop at various times after the diagnosis of SLE. It has been documented that the appearance of serum autoAbs [209] and pro-inflammatory cytokines [210] precedes the diagnosis of SLE by months to several years. Therefore, it is plausible to assume that certain biological processes that result in the production of proteins or other molecules occur in the serum or urine prior to the clinical diagnosis of LN. Studies identifying biomarkers for predicting an upcoming renal involvement in LN are lacking. Although our group has shown that IgG from patients with LN, but not from patients with SLE without LN, can injure cultured podocytes [211], it is still unclear when “podocytopathic” IgG first appears in the sera of patients with SLE prior to the development of LN. When injured, podocytes and other parenchymal cells are released into the urine. A more careful study of urine cells for the identification of molecules involved in kidney cell injury [211] can prove useful in monitoring the response to treatment, as the shed cells may provide a window into the events in the kidney tissue itself.

Several reasons may account for the fact that none of the discussed biomarkers has reached the bedside. There are challenges in both the discovery and clinical phase of biomarker validation [161]. During the discovery phases a major weight is placed on the formulation of clinically relevant questions and the identification of markers closely linked to the pathogenesis of LN, while less effort is made to consider the clinical and pathogenetic heterogeneity of LN in long prospective studies. In the efforts to validate the reported putative biomarkers, problems arise from the geographical and ethnic heterogeneity of the disease, the lack of standardized treatment protocols, sample collection and processing, and the methods used to measure various biomarkers.

## Figures and Tables

**Table 1 ijms-25-00805-t001:** Summary of biomarkers for lupus nephritis (LN).

1. Antibodies/immunoglobulins			
Sample	Biomarker	Association	Reference
Serum/plasma	Anti-dsDNA	Diagnosis, clinical disease activity, damage, and responses to therapy in LN	[40,106,109]
	Anti-C1q	Diagnosis, clinical disease activity, histological disease activity, and prognosis	[36,162,163,164,165,166]
	Anti-CRP	Clinical disease activity and responses to therapy in LN	[34,35]
	Anti-ENO-1 (+)	Diagnosis and prediction of LN	[37,38,40]
	AaA (Low)	Diagnosis	[41]
	Anti-chromatin	Diagnostic/predictive capacity in LN	[48]
	PTEC-binding IgG (+)	Clinical disease activity	[167]
	PHACTR4 icx (+)	Diagnosis	[168]
	P3H1 icx (+)	Diagnosis	[168]
	RGS12 icx (+)	Diagnosis	[168]
	PTEC-binding IgG (+)	Clinical disease activity	[167]
	IgM (↑)	Responses to therapy in LN	[169]
	ANCAs (+)	Prognosis	[32,49,50]
2. Kidney disease-related			
Serum	Hyperuricemia	Diagnosis	[78,170,171]
	Creatinine (↑)	Diagnosis and prognostic biomarkers	[78,166,172]
	Urea (↑)	Diagnosis and damage (>10.25 mmol/L)	[78]
Urine	Albumin to globulin ratio (low)	Diagnosis	[173]
	Proteinuria (↑) (>500 mg/24 h)	Diagnosis, clinical disease activity, histological disease activity, and prognosis	[67,95,109,174]
	Proteinuria (↓)	Responses to therapy in LN	[169]
	uPCR (↓) (<1.5 g/g at month 6)	Responses to therapy in LN	[109]
	WBC (↑)	Clinical disease activity	[67]
	RBC (↑)	Clinical disease activity	[67]
	Granular casts (+)	Clinical disease activity	[67]
3. Complement/Lymphocytes			
Serum	C3 (low)	Diagnosis, clinical disease activity, histological disease activity, responses to therapy in LN, and prognosis	[67,95,109,174]
	C4 (low)	Diagnosis and clinical disease activity	[175]
	C1q (low)	Histological disease activity	[165]
	Lymphocyte count (↑)	Responses to therapy in LN	[169]
4. Cytokines			
Serum	TWEAK	Diagnosis	[87,105,176,177,178]
	IL-2R (↓)	Responses to therapy in LN	[179]
	IL-8 (↓)	Responses to therapy in LN	[179]
	IL-10 (↑)	Clinical disease activity	[67,175]
	IL-17	Clinical disease activity and histological disease activity	[85,180]
	IL-23 (↓)	Responses to therapy in LN	[85]
Urine	TWEAK	Diagnosis and clinical disease activity	[87,176,177]
	TGF-β1 (↑)	Clinical disease activity and histological disease activity	[70,73]
	IL-17 (↑)	Diagnostic potential and clinical disease activity	[89]
	IL-12p40 (↑)	Diagnostic potential and clinical disease activity	[89]
	IL-15	Diagnostic potential and clinical disease activity	[89]
	IL-16 (↑)	Histological disease activity	[70]
	TARC (↑)	Diagnostic potential and clinical disease activity	[89]
	PF-4 (↑)	Clinical disease activity	[72]
5. Chemokines/Cell adhesion molecules			
Serum	APRIL (↑)	Predictive of treatment failure at 6 months	[61]
	BAFF (↓)	Predictive of clinical and histological responses to therapy in LN	[59]
	VCAM-1 (↑)	Clinical disease activity	[103]
	OPG (↓)	Responses to therapy in LN	[179]
Urine	APRIL (↑)	Diagnosis	[53,181]
	BAFF (↑)	Diagnosis	[53,181]
	CXCL4 (↑)	Diagnosis	[98]
	MCP-1 (↑)	Diagnosis, clinical disease activity, histological disease activity (proliferative vs. membranous), and responses to therapy in LN	[79,80,83,99,177,182,183]
	ALCAM (↑)	Diagnosis, clinical disease activity, histological disease activity, and prognosis	[94,95,184]
	VCAM-1 (↑)	Diagnosis, clinical disease activity, histological disease activity (proliferative vs. membranous), damage, and prognostic biomarkers	[72,75,94,98,99]
	ICAM-1 (↑)	Clinical disease activity	[97]
	NCAM-1 (↑)	Clinical disease activity	[97]
	IP-10/CXCL10 (↑)	Diagnostic potential and clinical disease activity (renal)	[67,89]
6. Other proteins			
Serum	Axl (↑)	Diagnosis, clinical disease activity, responses to therapy in LN, and prognostic biomarkers	[106,107,108]
	HE4 (↑)	Diagnosis	[78,185]
	IGFBP-2 (↑)	Diagnosis, clinical disease activity, and damage	[186]
	IGFBP-4	Damage	[187]
	sTNFRII (↑)	Diagnosis, clinical disease activity, histological disease activity, damage, responses to therapy in LN, and prognosis	[106,188,189]
	Angiostatin (↑)	Clinical disease activity	[108]
	Ferritin (↑)	Clinical disease activity	[108]
	Progranulin (↑)	Clinical disease activity	[108]
	SDC-1 (↑)	Clinical disease activity and histological disease activity	[103]
	Resistin (↑)	Damage	[190]
	CSF-1 (↓)	Responses to therapy in LN	[191]
	HNP1-3 (↓)	Responses to therapy in LN	[192]
	S100A8/A9 (↑)	Responses to therapy in LN	[193]
	S100A12 (↑)	Responses to therapy in LN	[193]
Urine	Angiostatin	Diagnosis, clinical disease activity, histological disease activity, and damage	[51,74,75,98]
	NGAL (↑)	Diagnosis, clinical disease activity, and (↓) responses to therapy in LN	[80,104,162,194,195,196,197,198,199]
	TF (↑)	Diagnosis, clinical disease activity and responses to therapy in LN	[92,183,200]
	β2-MG (↑)	Diagnosis	[22,201]
	Angptl4 (↑)	Clinical disease activity	[51,73]
	Calpastatin (↑)	Clinical disease activity	[72]
	CD163 (↑)	Clinical disease activity, histological disease activity (predictor, proliferative vs. non-proliferative), (↓) responses to therapy in LN and prognosis	[70,109,110,182]
	FOLR2 (↑)	Clinical disease activity	[73]
	Hemopexin (↑)	Clinical disease activity	[72]
	L-selectin (↑)	Clinical disease activity	[73]
	PDGFRβ (↑)	Clinical disease activity	[73]
	Peroxiredoxin 6 (↑)	Clinical disease activity	[72]
	Progranulin (↑)	Clinical disease activity	[108]
	Properdin (↑)	Clinical disease activity	[72]
	RBP4 (↑)	Clinical disease activity and (↓) responses to therapy in LN	[202]
	TSP1 (↑)	Clinical disease activity	[73]
	TTP1 (↑)	Clinical disease activity	[73]
	NRP-1 (↑)	Responses to therapy in LN	[203]
	Plasmin (↑)	Clinical disease activity	[200]
	TFPI (↑)	Clinical disease activity	[200]
	EGF (↓)	Prognosis	[112]
7. MicroRNAs (miRNAs)			
Serum/plasma	miRNA-21	Diagnosis	[128,204]
Urine	miRNA-31-5p (↑)	Responses to therapy in LN	[127]
	miRNA-107 (↑)	Responses to therapy in LN	[127]
	miRNA-135b-5p (↑)	Responses to therapy in LN	[127]
8. Microparticles (MP)			
Urine	MP-CX3CR1+ (↑)	Diagnosis	[172]
	MP-HLADR+ (↑)	Diagnosis	[172]
	MP-HMGB1+ (↑)	Diagnosis and clinical disease activity (active vs. non-active)	[172]
9. Renal tissue			
Kidney biopsy	Mannose-enriched N-glycan expression	Diagnosis and prognosis	[205]
	CSF-1 (↑)	Histological disease activity	[191]
	Periostin (↑)	Damage	[206]
	C9 (+)	Prognosis	[207]
	Podocyte foot process width (↓)	Prognosis	[169]
	Arteriolar C4d deposition (+)	Prognosis	[208]
	Cellular crescents (+)	Prognosis	[172]
	Fibrous crescents (+)	Prognosis	[172]
	Glomerular C3 deposition (+)	Prognosis	[212]
	IFTA (+) (≥25% of the surface cortical area)	Prognosis	[213]
	Vascular injury (+) (≥25% subintimal narrowing of the lumen)	Prognosis	[213]

AaA: Anti-actin antibody; ANCA: Anti-neutrophil cytoplasmic antibody; Ang2: Angiopoietin 2; Angptl4: Angiopoietin-like 4; Anti-C1q: Anti-complement component 1q; Anti-CRP: Anti-C-reactive protein; Anti-dsDNA: Anti-double-stranded deoxyribonucleic acid; Anti-ENO-1: Anti-Enolase 1; APRIL: A proliferation-inducing ligand; BAFF: B-cell activating factor; β2-MG: Beta-2 microglobulin; C1q: Complement component 1q; C3: Complement component 3; C4: Complement component 4; C4d: Complement component 4d; C9: Complement component 9; Cer: Ceramide; CSF-1: Colony stimulating factor 1; CXCL4: C-X-C motif chemokine ligand 4; EGF: Epidermal growth factor; FOLR2: Folate receptor beta; HNP1-3: Human neutrophil peptide 1-3; ICAM-1: Intercellular adhesion molecule 1; IFTA: Interstitial fibrosis and tubular atrophy; IGFBP: Insulin-like growth factor-binding protein; IL: Interleukin; IL-2R: Interleukin 2 receptor; IL-12p40: Interleukin 12 subunit p40; IP-10: Interferon gamma-induced protein 10, also known as CXCL10; MCP-1: Monocyte chemoattractant protein 1; MicroRNAs (miRNAs): Micro-ribonucleic acids, with miRNA-21 being Micro-ribonucleic acid 21; MP-CX3CR1+: Microparticles with CX3CR1 expression; MP-HLADR+: Microparticles with HLA-DR expression; MP-HMGB1+: Microparticles with high mobility group box 1 protein expression; NRP-1: Neuropilin 1; NCAM-1: Neural cell adhesion molecule 1; NGAL: Neutrophil gelatinase-associated lipocalin; OPG: Osteoprotegerin; PDGFRβ: Platelet-derived growth factor receptor beta; PF-4: Platelet factor 4; PHACTR4: Phosphatase and actin regulator 4, with PHACTR4 icx being Phosphatase and actin regulator 4 immune complexes; P3H1: Prolyl 3-hydroxylase 1, with P3H1 icx as Prolyl 3-hydroxylase 1 immune complexes; PTEC-binding IgG: Proximal tubular epithelial cell-binding immunoglobulin G; RBC: Red blood cells; RGS12: Regulator of G-protein signalling 12, with RGS12 icx as Regulator of G-protein signalling 12 immune complexes; SDC-1: Syndecan-1; sTNFRII: Soluble tumor necrosis factor receptor II; TARC: Thymus and activation-regulated chemokine, also known as CCL17; TFPI: Tissue factor pathway inhibitor; TGF-β1: Transforming growth factor-beta 1; TSP1: Thrombospondin 1; TTP1: Tripeptidyl peptidase 1; Type I IFN: Type I Interferon; TWEAK: Tumor necrosis factor-like weak inducer of apoptosis; uPCR: Urine protein to creatinine ratio; VCAM-1: Vascular cell adhesion molecule 1; WBC: White blood cells; (+): Positivity; ↓: Decreased; ↑: Increased.

## Data Availability

Not applicable.

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
