# Peer review of "Lupus Nephritis Biomarkers: A Critical Review"

_ijms, 2024, doi:10.3390/ijms25020805_

Round 1
Reviewer 1 Report
Comments and Suggestions for Authors
He reviewed the manuscript titled “Lupus Nephritis Biomarkers: A Critical Review.” It is an interesting, well-written manuscript that covers the most prominent markers in the different tissues and fluids of the body. From my point of view, the authors present a complete and well-structured work, however there are some points that should be reviewed:
1. In point 2.2 Cytokines/chemokines. Due to the importance of chemokines, I suggest that the authors not only expand the text referring to these molecules, but also give examples of them, since they promise to be good markers in the diagnosis and prognosis of LN.
2. In section 2.7. Genetics and Epigenetics. I suggest that these two items be mentioned separately, since although they are related, they are totally different questions and topics.
3. In section 2.7. Referring to the topic of "Genetics" the researchers could talk and include variants of a single nucleotide specific for LN, since the researchers refer to reports of patients with systemic lupus erythematosus in which monogenic defects have been characterized, however they do not give any example of them, so it would be advisable to make a table indicating the gene and SNVs involved in LN.
4. I suggest researchers add point 2.8 in which only epigenetics is possible. In this area, researchers could mention those genes that have been reported as altered in their methylation pattern and that have been related to autoimmune diseases and specifically to LN.
Comments on the Quality of English LanguageI suggest minimal revision of writing in English.
Author Response
We thank the Editor and reviewers for their insightful comments. We have made corrections to our manuscript accordingly. Please find below our point-by-point responses to each comment.
Comments to the Author
He reviewed the manuscript titled “Lupus Nephritis Biomarkers: A Critical Review.” It is an interesting, well-written manuscript that covers the most prominent markers in the different tissues and fluids of the body. From my point of view, the authors present a complete and well-structured work, however there are some points that should be reviewed:
- In point 2.2 Cytokines/chemokines. Due to the importance of chemokines, I suggest that the authors not only expand the text referring to these molecules, but also give examples of them, since they promise to be good markers in the diagnosis and prognosis of LN.
Response: Thank you for your invaluable comments and insightful suggestions. We have checked this, as suggested, and made changes as needed; they are highlighted in section 2.2 Cytokines/Chemokines (pages 3-4, lines 143–152).
“The deposition of ICs on glomeruli initiates and activates the complement pathway, attracting immune cells and triggering the release of inflammatory cytokines, thereby exacerbating glomerular damage. For instance, B-cell activating factor (BAFF or BLyS), produced by glomerular macrophages and mesangial cells, stimulates B-cell activation both directly and by inducing the production of pro-inflammatory molecules, such as IFN-α, perpetuating a detrimental cycle in the renal microenvironment. Moreover, cytokines, such as urinary monocyte chemoattractant protein-1 (MCP-1/CCL2), BAFF, and TNF-related weak inducer of apoptosis (TWEAK), are pertinent in predicting active LN, assessing therapeutic responses, and forecasting disease flares [55, 56].”
- In section 2.7. Genetics and Epigenetics. I suggest that these two items be mentioned separately, since although they are related, they are totally different questions and topics.
Response: We have checked this, as suggested, and presented them in two separate sections 2.7 and 2.8 (pages 7 and 9, lines 351 and 418).
- In section 2.7. Referring to the topic of "Genetics" the researchers could talk and include variants of a single nucleotide specific for LN, since the researchers refer to reports of patients with systemic lupus erythematosus in which monogenic defects have been characterized, however they do not give any example of them, so it would be advisable to make a table indicating the gene and SNVs involved in LN.
Response: We have reviewed this suggestion and revised accordingly the section to include information on the high prevalence of LN in these patients. We have also added examples of SNVs associated with LN. These updates can be found in Section 2.7, on pages 7-8 , lines 355-357, 363-399 and 363-389 and 397-399.
“ One to three percent of patients with SLE may have a single-gene (monogenic) defect, and these patients experience a high incidence of LN [139-141]. ”
“For example, recent advances in whole-exome sequencing have revealed novel mutations in the TNF alpha induced protein 3 (TNFAIP3) gene to be linked to LN. These mutations lead to increased activity in nuclear factor kappa B and type I IFN pathways, along with a rise in pro-inflammatory cytokines [143]. Specifically, this study linked three novel mutations with LN: c.634+2T>C, exon 7–8 deletion, and c.1300_1301delinsTA (p. A434*) [143]. Additionally, various databases have reported TNFAIP3 mutations associated with kidney involvement, such as the p.Q187* mutation in a patient with proliferative LN and the p.F224Sfs*4 mutation in a patient with membranous LN [143-145]. Moreover, a genome-wide association study focusing on female SLE patients of European ancestry with LN has identified genes, such as FCGR, STAT4, and BANK1. These genes have been shown to correlate with both the occurrence and severity of LN, which has been confirmed across various cohorts [146-150]. Wang et al. used weighted gene co-expression network analysis to identify four hub genes, namely CD53 (AUC = 0.995), TGFBI (AUC = 0.997), MS4A6A (AUC = 0.994), and HERC6 (AUC = 0.999), which are involved in the development of the inflammatory response and immune activation in LN (p < 0.0001) [151]. Yavuz et al. discovered that MERTK, a novel genetic region, contributes to the risk of developing LN and ESRD in patients with SLE, a finding that has been replicated across various ethnicities [152]. MERTK, a member of the Tyro3/Axl/Mer receptor kinase family, is the primary receptor on macrophages for apoptotic cells [153, 154]. It plays a key role in the regulation of the innate immune response through efferocytosis, notably influencing the production of cytokines, such as IL-10, TGF-β, IL-6, and IL-12 [155, 156]. Furthermore, MERTK is critical in suppressing TLR-mediated innate immune responses by activating STAT1, which leads to an anti-inflammatory feedback through the production of the cytokine signaling suppressors SOCS1 and SOCS3 [157]. Other genes, such as PRDM1, are associated with proliferative LN [152] and play a vital role as modulators of dendritic cell function and repressors of the IFN-β gene [158].”
“Additionally, human leukocyte antigen (HLA) variants have long been found to be linked to the development of LN, with HLA-DR3 and HLA-DR15 increasing the risk significantly [159]. ”
- I suggest researchers add point 2.8 in which only epigenetics is possible. In this area, researchers could mention those genes that have been reported as altered in their methylation pattern and that have been related to autoimmune diseases and specifically to LN
Response: We have addressed this point and as suggested, we made changes as needed; they are highlighted in section 2.8 (page 9, lines 423–424).
“For example, alterations in DNA methylation of the MERTK gene may modify its activity and increase the risk of SLE-ESRD [152]”
- (x) Minor editing of English language required
Response: We have had the revised manuscript re-edited for language and grammar.

Reviewer 2 Report
Comments and Suggestions for Authors
This is a comprehensive review of the different biomarkers which have been tested in the context of lupus nephritis. Therefore it can serve as a useful resource. I have very few comments.
1. Serum autoantibodies. I miss ANCA/anti-MPO which have been associated with severe presentations/prognosis of lupus nephritis.
2. Also, what about anti-histone/chromatin autoantibodies? There have been reports about their predictive capacity in lupus nephritis.
3. Genetics. Could the authors briefly discuss the possible utility of polygenic risk scores in classifying patients according to their risk for nephritis?
4. Can the authors discuss what might be the reasons that none of these biomarkers has actually reached to bedside? Is it the lack of standardisation? Insufficient validation? Lack of independent predictive value.
Author Response
We thank the Editor and reviewers for their insightful comments. We have made corrections to our manuscript accordingly. Please find below our point-by-point responses to each comment.
Reviewer: 2
This is a comprehensive review of the different biomarkers which have been tested in the context of lupus nephritis. Therefore it can serve as a useful resource. I have very few comments.
- Serum autoantibodies. I miss ANCA/anti-MPO which have been associated with severe presentations/prognosis of lupus nephritis.
Response: Thank you for your meticulous review and pertinent comments. This was mentioned in Table 1 under Antibodies/immunoglobulins and has now been further expanded in the revised section 2.1. serum autoAbs (page 3, lines 128–137).
“Interestingly, numerous studies have reported that patients with LN and antineutrophil cytoplasmic antibodies (ANCA), specifically those who are anti-myeloperoxidase (MPO)/proteinase 3 (PR3)-positive or double positive for anti-MPO and anti-PR3 antibodies, exhibit serologically active SLE (higher dsDNA antibody titers and lower serum C4 concentrations) compared with ANCA-negative LN patients. Additionally, patients in the subgroup with proliferative LN display more necrosis in renal biopsy specimens than ANCA-negative patients. However, it is noteworthy that no significant difference was reported between the outcomes of these groups [50-52]. Yet, given the poor prognosis of ANCA-positive cases, it is suggested that all patients with LN be screened for ANCA.”
- Also, what about anti-histone/chromatin autoantibodies? There have been reports about their predictive capacity in lupus nephritis.
Response: We have checked this, as suggested, and made corrections where needed. They are highlighted in section 2.1. serum autoAbs (page 3, lines 121–127).
“Histones are a primary component of the nucleosome, and studies have indicated a strong correlation between antibodies against chromatin/nucleosomes and LN as well as an increased risk of developing proliferative LN [43, 44]. Histone 1 forms part of the apoptotic chromatin constituents in the dense electron deposits of the glomerular basement, which are recognized targets of nephritogenic dsDNA antibodies [45-48]. Antibodies to chromatin are present in patients with LN but not particularly useful in monitoring disease activity of response to treatment [49].”
- Could the authors briefly discuss the possible utility of polygenic risk scores in classifying patients according to their risk for nephritis?
Response: We have checked this, as suggested, and made corrections where needed; they are highlighted in section 2.7 (pages 8-9, lines 400–416).
“In terms of classifying patients based on their risk of nephritis, Chen et al. observed that a high polygenic risk score for SLE correlates with poorer prognostic factors like earlier age-of-onset and LN [163]. Kwon et al. reported in a Korean cohort that an individual’s highest weighted genetic risk score (wGRS), calculated from 112 well-validated non-HLA single nucleotide polymorphisms and HLA haplotypes of SLE-risk loci, was independently associated with the development of LN and the production of anti-Sm antibody, compared to those in the lowest wGRS quartile [164]. This association was observed regardless of the age of onset [164]. Webber et al. observed both HLA and non-HLA SLE risk-weighted genetic risk scores were significantly associated with the risk of proliferative LN in two large, multi-ethnic cohorts of 1251 SLE patients [165]. This may indicate that SLE risk loci are of greater importance in the development of proliferative LN, as opposed to non-proliferative LN [165]. Despite these developments, the full spectrum of genetic risk factors for LN and its progression to ESRD remains to be better understood and further developed. Taken together, a gene expression score, including all variants known to contribute to the development of LN (class, severity, and risk of ESRD), can be used to screen all patients with SLE, identify patients at high risk of LN, and predict various clinical outcomes in SLE patients.”
- Can the authors discuss what might be the reasons that none of these biomarkers has actually reached to bedside? Is it the lack of standardisation? Insufficient validation? Lack of independent predictive value.
Response: We agree with your valuable comment and have made the necessary corrections which are highlighted in the Conclusions and Future Directions (page 9, lines 453–461).
“ Several reasons may account for the fact that none of the discussed biomarkers has reached the bedside. There are challenges in both the discovery and clinical phase of biomarker validation [171]. During the discovery phases a major weight is place in the formulation of clinically relevant questions and the identification of markers closely linked to the pathogenesis of LN, while less effort is made to consider the clinical and pathogenetic heterogeneity of LN in long prospective studies. In efforts to validate reported putative biomarkers, problems arise from the geographical and ethnic heterogeneity of the disease, the lack of standardized treatment protocols, sample collection and processing and methods of measuring various biomarkers. ”
- (x) English language fine. No issues detected
Response: Thank you for your meticulous review of our paper.

Reviewer 3 Report
Comments and Suggestions for Authors
The evaluated article is a review type that explores current knowledge regarding possible biomarkers associated with lupus nephritis. The subject is very important and actual in the modern rheumatology and the article evaluates several interesting aspects. Both the abstract and the introductory part are relatively shorter than expected. Due to the fact that it is possible that not all readers are aware of SLE, there should also be some data related to lupus and the positioning of lupus nephritis as an important complication.
Also, the established types of lupus nephritis are not mentioned and, consequently, there is no discussion related to the possibility that certain biomarkers correlate with certain types of nephritis.
The discussed biomarkers are exemplified in different chapters and paragraphs with the mention of the statistical significance (p) from various studies, which loads the text quite a lot. One option would be to enter the respective data in the table (study, number of participants, statistical significance, result) and keep only the result in the text.
Author Response
We thank the Editor and reviewers for their insightful comments. We have made corrections to our manuscript accordingly. Please find below our point-by-point responses to each comment.
Reviewer: 3
Comments to the Author
1. The evaluated article is a review type that explores current knowledge regarding possible biomarkers associated with lupus nephritis. The subject is very important and actual in the modern rheumatology and the article evaluates several interesting aspects. Both the abstract and the introductory part are relatively shorter than expected. Due to the fact that it is possible that not all readers are aware of SLE, there should also be some data related to lupus and the positioning of lupus nephritis as an important complication.
Response: We agree with your pertinent comment and have made the necessary corrections, as highlighted in the abstract and introduction (page 1, lines 12-14, 17-19 and 25-37)
Abstract
“ Despite marked improvements in the survival of patients with severe LN over the past 50 years, complete clinical remission after immunosuppressive therapy is achieved in only half of the patients. ”
“ Renal biopsy remains the gold standard for the identification of histological phenotypes of LN which guides disease management. However, the molecular pathophysiology of specific renal lesions remains poorly understood.”
Introduction
“Systemic lupus erythematosus (SLE) is a chronic systemic autoimmune disease characterized by the presence of autoantibodies (autoAbs), autoreactive B and T cells, and dysregulation of cytokines, which lead to inflammation and damage multiple organs [1-4]. The prevalence of SLE in the United States ranges from 20 to 150 cases per 100,000 people [5-8]. The etiology and pathogenesis of SLE are not well understood, the factors that lead to disease onset are highly variable, and the disease manifests systemically with manifestations resulting from the injury of multiple tissues. The kidney is the most commonly involved organ in this disease, extensively contributing to morbidity and mortality [9-11].
Lupus nephritis (LN) has been classified histologically in six types, which are determined by the location and the type of histological changes. There is variability among the six classes in terms of response to treatment and preservation of the kidney function and the development of end stage disease [12]. ”
- Also, the established types of lupus nephritis are not mentioned and, consequently, there is no discussion related to the possibility that certain biomarkers correlate with certain types of nephritis.
Response: We have revised the introduction to encompass information about the established types of lupus nephritis. Moreover, the role of specific biomarkers, including transcriptome analysis, antibodies, cytokines, and others, has been thoroughly discussed in the respective sections across the manuscript. This discussion highlights their significance in distinguishing each class of lupus nephritis. The pertinent details can be found on pages 2, 4 and 6, lines 69-73, 90-92, 153-162 and 261-269 .
“ For instance, the signature of type I interferon (IFN) is predominantly observed in the kidneys and skin of individuals diagnosed with proliferative LN [3, 24, 25], whereas patients with membranous LN exhibit different transcriptomic patterns [25], and transcriptome-based investigations using animal models have provided insights into the progression and phases of LN development [26]. ”
“ Associations between specific autoAbs, including anti-double stranded DNA (dsDNA) and anti-C1q antibodies, and distinct histological classifications of LN have been reported [27-29]. ”
“ Furthermore, distinct cytokines predominate in different types of LN. For example, in proliferative LN (class III/IV), the deposition of ICs beneath the endothelium leads to mesangial cell proliferation, an increase in extracellular matrix production, and the release of a remarkable array of pro-inflammatory cytokines, such as type I IFN, interleukin (IL)-1β, IL6, IL8, IL-37, and IL-17A [58]. Conversely, membranous LN (class V) is characterized by subepithelial localization of ICs, resulting in increased complement activation and reduced inflammatory responses [58]. Levels of cytokines may prove useful in distinguishing between proliferative and non-proliferative forms of LN and this information may be useful in guiding precise treatment and management strategies and may eliminate the need for invasive LN biopsy [58]. ”
“ Moreover, a combination of urine markers (uVCAM-1, uCystatinC, and uKIM-1) yielded a promising AUC of 0.80 (95% CI: 0.69–0.90), highlighting the potential to differentiate between the proliferative and membranous forms of LN [54]. Levels of uCAMs collectively serve as robust tools in evaluating LN activity and informing clinical decisions [106]. Urine ALCAM (uALCAM) levels have been suggested to be more efficient in distinguishing proliferative LN from membranous LN [98]. Moreover, uALCAM and urine VCAM-1 (uVCAM-1) levels exhibit a strong correlation with renal histological activity, underscoring their potential value as LN activity markers [78, 83, 98, 101]. ”
The discussed biomarkers are exemplified in different chapters and paragraphs with the mention of the statistical significance (p) from various studies, which loads the text quite a lot. One option would be to enter the respective data in the table (study, number of participants, statistical significance, result) and keep only the result in the text.
Response: We agree with your suggestion to add the p-value in the existing table. However, as the table presents LN biomarker data from various studies, summarizing the role of each biomarker in diagnosis, flare detection, and response to therapy, including the statistical significance (p-value) in the table could lead to overcrowding. An alternative option is to delete the p-value from the text.

Round 2
Reviewer 3 Report
Comments and Suggestions for Authors
New version satisfies all my previous recommendations.